# Effects of C-Terminal Lys-Arg Residue of AapA1 Protein on Toxicity and Structural Mechanism

**DOI:** 10.3390/toxins15090542

**Published:** 2023-09-02

**Authors:** Zanxia Cao, Liling Zhao, Tingting Yan, Lei Liu

**Affiliations:** 1Shandong Provincial Key Laboratory of Biophysics, Institute of Biophysics, Dezhou University, Dezhou 253023, China; zhaollabc@163.com (L.Z.); yantt2023@163.com (T.Y.); 2College of Physics and Electronic Information, Dezhou University, Dezhou 253023, China

**Keywords:** AapA1 membrane, toxin–antitoxin (TA) system, molecular dynamics simulation, C-terminal charged residues, structural characteristics

## Abstract

Previous experimental investigations have established the indispensability of the C-terminal Lys-Arg residues in the toxic activity of the AapA1 toxin protein. AapA1 is classified as a type I toxin–antitoxin (TA) bacterial toxin, and the precise impact of the C-terminal Lys-Arg residues on its structure and mechanism of action remains elusive. To address this knowledge gap, the present study employed molecular dynamics (MD) and enhanced sampling Well-tempered Two-dimensional Metadynamics (2D-MetaD) simulations to examine the behavior of the C-terminal Lys-Arg residues of truncated AapA1 toxin (AapA1-28) within the inner membrane of *Escherichia coli*. Specifically, the study focused on the elucidation of possible conformation states of AapA1-28 protein in POPE/POPG (3:1) bilayers and their interactions between the protein and POPE/POPG (3:1) bilayers. The findings of our investigation indicate that the AapA1-28 protein does not adopt a vertical orientation upon membrane insertion; rather, it assumes an angled conformation, with the side chain of Lys-23 directed toward the upper layer of the membrane. This non-transmembrane conformation of AapA1-28 protein impedes its ability to form pores within the membrane, resulting in reduced toxicity towards *Escherichia coli*. These results suggest that C-Terminal positively charged residues are essential for electrostatic binding to the negatively charged head group of bottom bilayer membrane, which stabilize the transmembrane conformation. These outcomes contribute to our comprehension of the impact of C-terminal charged residues on the structure and functionality of membrane-associated proteins, and provide an improved understanding of how protein sequence influences the antimicrobial effect.

## 1. Introduction

The locus *aapA1*/IsoA1 [1,2] of type I toxin–antitoxin (TA) system [3] is expressed from the chromosome of the human pathogen *Helicobacter pylori (H. pylori)*. The *aapA1* gene encodes the toxin protein AapA1, a small hydrophobic protein of 30 amino acids. AapA1 protein is responsible for targeting the inner membrane of bacteria, and ectopic overexpression can lead to growth arrest or cell death. Experimental evidence suggests that AapA1 inserts vertically into membrane with a stable transmembrane alpha-helical conformation (PDB id: 6GIG) [4]. For *H. pylori*, the ectopic overexpression of AapA1 protein leads to a growth arrest and induces a morphological transformation from spiral to coccoid [5]. For *Escherichia coli (E. coli)* [4], OD600 growth curves indicated that the overexpression of the full-length AapA1 protein had an obvious toxic effect. For most membrane-associated type I toxin protein, the C-terminal part has positively charged residues. These positively charged residues were shown to be important for the binding of toxin protein on negatively charged bacterial membrane. In order to assess the impact of C-terminal residues on the toxicity of AapA1, Korkut et al. [4] produced specific constructs expressing truncated variant of the toxin that did not contain the last two Lys-Arg (KR) residues (named as AapA1-28 protein). The toxic effect of AapA1-28 protein was evaluated in liquid growth media after over-expression of this protein during growth of *E. coli.* While AapA1 displays the toxicity effect, the AapA1-28 protein does not exhibit any toxic effect on *E. coli* growth. Meanwhile, the alignment of 166 AapA proteins of 91 *H. pylori* strains revealed a remarkable conservation of C-terminal KR residues. These findings highlight the crucial role played by the C-terminal KR residues in the toxin’s toxicity. Consequently, it is imperative to analyze the structural characteristics of AapA1-28 within the *E. coli* inner membrane and compare them with those of the full-length AapA1 toxin protein.

Plasmon waveguide resonance (PWR) and nuclear magnetic resonance (NMR) findings provide evidence that the AapA1 toxin protein exists in a transmembrane state within the membrane [4]. The considerable impact of the C-terminal KR residues on *E. coli* toxicity remains challenging to comprehend. Understanding the dynamic structural changes and processes of small toxin proteins within bilayer membranes is crucial for elucidating their functions. Molecular dynamics (MD) simulations in conjunction with enhanced sampling method provide an effective tool for revealing protein–membrane interactions at the atomistic level and complementing experimental results [6,7,8,9]. The first category of studies [4,10,11,12,13,14,15] typically involves placing monomeric or oligomeric toxin proteins/antibacterial peptides either on the surface of bilayer membrane or directly in a transmembrane helical state, with a focus on investigating local structural characteristics such as changes in membrane properties and peptide conformations via conventional molecular dynamics (CMD) simulations. Steinbrecher et al. [10] studied the protein–lipid interaction and structures of monomeric, dimeric, and oligomeric TisB toxin proteins by using coarse-grained MD simulation. The simulation results indicated that the transmembrane state is energetically feasible, the transmembrane dimers are stable, and the TisB oligomers that assembled as transmembrane pores are unstable. Schneider et al. [11] revealed the tetrameric charge-zipper assemble of the TisB toxin protein in membrane by using all-atom and coarse-grained MD simulations. The tetrameric structure is decribed as an antiparallel dimer-of dimers. Leveritt et al. [12] compared the structural characteristics of melittin tetramer in DMPC and DMPC/DMPG membrane. Stable pore can be formed in DMPC, but the pore is not stable in DMPC/DMPG membrane. However, due to limited sampling efficiency and time, these CMD simulations face challenges in directly observing various free energy minima and potential pore structures of toxin proteins/antibacterial peptides within cell membranes. The second category of studies [16,17,18,19,20,21,22] examines the global structural characteristics of the toxin protein/antibacterial peptide–cell membrane system using enhanced sampling method. These simulations typically employ techniques such as umbrella sampling, replica exchange umbrella sampling, or various other Metadynamics methods to explore the free energy surface of protein/peptide–cell membrane system. Yeasmin et al. [16] investigated the interaction mechanism of human β defensing type 3 antimicrobial peptide with membrane by using umbrella sampling simulations. The simulation result suggests a toroidal pore model for this antimicrobial peptide. Cao et al. [20] investigated the translocation mechanism and structural characteristics of five spontaneous membrane translocating peptides by using bias-exchange Metadynamics simulations. Simulations results yield sequence-dependent free energy barrier and translocation process. A critical step in these simulations is the selection of appropriate collective variables (CVs) to effectively capture the free energy surface, as the suitable CVs are not known a priori. Hub et al. [21] presented good reaction coordinates for describing the pore formation of membrane-active peptides.

Taking into account the aforementioned considerations, the present study aimed to examine the consequences of removing C-terminal KR residues on the structure and function of AapA1 toxin in POPE/POPG bilayer membrane. Well-tempered Two-dimensional Metadynamics (2D-MetaD) simulations were specifically employed to calculate the conformational free energy differences between two distinct states and identify representative conformations. In parallel, microsecond conventional molecular dynamics (CMD) simulations were conducted to explore the dynamic changes in structure and interactions of the AapA1-28 protein with model *E. coli* membranes. The POPE/POPG (3:1) lipid bilayer served as a representative model of the *E. coli* inner membrane, enabling comparisons of the structural characteristics among different protein–membrane systems. These findings lay a foundation for advancing the design of antimicrobial agents.

## 2. Results

### 2.1. Structural and Positional Changes of Tetrameric AapA1-28 Protein Elucidated through CMD_T1/2 Simulations

PWR and NMR studies on AapA1 protein have indicated that the protein adopts a vertical transmembrane state [4]. The C-terminal charged residues play a critical role in the protein’s toxicity, as their removal reduces its membrane-destructive ability. However, the underlying reasons for this effect remain unclear. Snapshots of the tetrameric AapA1-28 protein–membrane structure at different time points are displayed in Figure 1. Simulation details are shown in Table 1. In the initial conformation, the angle between the protein’s helical axis and the bilayer normal is less than 10°. The distance between the C-terminal five residues 24–28 (VVVLL) and the center of membrane is approximately −1 nm. K-11 and K-16 residues interact with the upper membrane, while K-23 residues interact with the bottom membrane. However, this initial transmembrane state is not entirely stable for a portion of the AapA1-28 protein. Three significant changes occur in the AapA1-28 protein: the angle between the protein’s helical axis and the bilayer normal increases (as shown in Appendix A), the distance between C-terminal five residues 24–28 (VVVLL) and the membrane center decreases (as shown in Appendix A), and the angle between K-23 side chain and membrane *Z*-axis (as shown in Figure 2) increases. K-23 side chain changes from a downward orientation to an upward orientation, interacting with the upper membrane.

In general, the monomeric AapA1-28 protein exhibits two distinct conformational states: the first conformation corresponds to the vertical transmembrane state where the K-23 residue interacts with the bottom membrane (as shown in Figure 3C) and the second conformation involves the K-23 residue interacting with the upper membrane (as shown in Figure 3D). Determining the preferred conformational state of the AapA1-28 protein in the POPE/POPG membrane is essential for further analysis and understanding its behavior and functional implications.

### 2.2. Two-Dimensional Free Energy Landscapes (FELs) from 2D-MetaD Simulations

To compare the free energy differences between the two distinct conformational states, we calculated the two-dimensional free energy (ΔG) as functions of CV1 and CV2 (as illustrated in Figure 4). The 2D-MetaD simulation details are shown in Table 1. For WT1, we identified two local energy minima: region A (CV1 = 65, CV2 = 35) with a free energy of 0 kJ/mol and region B (CV1 = 35, CV2 = 55) with a free energy of 30.6 kJ/mol. Similarly, for WT2, we located three local energy minima: region A’ (CV1 = 75, CV2 = 35) at 0 kJ/mol, region B’ (CV1 = 35, CV2 = 85) at 46.1 kJ/mol, and region C’ (CV1 = 85, CV2 = 85) at 37.4 kJ/mol. These local minima, labeled as A, B, A’, B’, and C’, are depicted in Figure 5 and served as the initial conformations for the CMD_M simulations. In general, these conformations can be categorized based on the structural characteristics of the K-23-membrane. In the first category, the AapA1-28 protein was inserted into the lipid bilayer membrane, aligning itself along the bilayer’s *Z*-axis, with the side chain of the K-23 residue pointing towards the bottom layer of membrane. In the second category, the AapA1-28 protein was inserted into the bilayer membrane at an angle with the side chain of the K-23 residue pointing towards the upper layer of membrane. Regardless of the initial conformation, the global minimum free energy conformation is characterized by the interaction between the K-23 residue with the upper membrane.

### 2.3. Interactions between the AapA1-28 Protein and Membrane

In addition to the 2D-MetaD simulations, we conducted five sets of 1000 ns CMD simulations of membrane systems incorporating AapA1-28, starting from the local minimum energy state obtained from the 2D-MetaD simulations (detailed information presented in Table 1). The binding free energies between AapA1-28 protein and the POPE/POPG (3:1) bilayer were evaluated for the last 200 ns of the simulations CMD_M (Table 2) using the g_mmpbsa [23]. The total binding free energy, considering the transmembrane region (residues 9–28) interaction with the membrane, is displayed in Table 2, while the binding free energy involving residues 1–28 and the membrane is provided in Appendix A. The calculation utilized Equation (1) along with the molecular mechanics Poisson Boltzmann surface area (MM-PBSA) method [23].
(1)$$∆G_{binding}=∆G_{elec}+∆G_{pols}+∆G_{vdW}+∆G_{npols}$$

Here, ΔG_elec_, ΔG_pols_, ΔG_vdw_, and ΔG_npols_ represent the electrostatic energy, polar solvation energy, van der Waals energy, and nonpolar solvation energy, respectively. Irrespective of whether K-23 interacts with the upper or bottom lipid membrane, no significant difference in binding free energy between the protein and the membrane was observed.

**Table 1 toxins-15-00542-t001:** Summary of molecular dynamics simulations.

Simulation Systems	Sampling Method	The Initial Conformation	Two Collective Variables (CVs)	Acronym	Simulation Length (ns)
Tetramer AapA1-28/POPE/POPG (3:1)	CMD simulation	Figure 1A		CMD_T1	5000 ns
		Figure 1B		CMD_T2	5000 ns
AapA1-28/POPE/POPG (3:1)	2D-MetaD simulation	Figure 3C	CV1: Contact number between K-23 and the top phosphate group. CV2: Contact number between K-23 and the bottom phosphate group.	WT1	2 replica× 1000 ns
AapA1-28/POPE/POPG (3:1)	2D-MetaD simulation	Figure 3D	The same CVs	WT2	2 replica ×1000 ns
AapA1-28/POPE/POPG (3:1)	CMD simulation	Figure 5A (representative conformation of region A in Figure 4)		CMD_M1	1000 ns
		Figure 5B (representative conformation of region B in Figure 4)		CMD_M2	1000 ns
		Figure 5C (representative conformation of region A’ in Figure 4)		CMD_M3	1000 ns
		Figure 5D (representative conformation of region B’ in Figure 4)		CMD_M4	1000 ns
		Figure 5E (representative conformation of region C’ in Figure 4)		CMD_M5	1000 ns

**Table 2 toxins-15-00542-t002:** The count of intermolecular interactions and the binding free energy (in kJ/mol) involving residues 9–28 of AapA1-28 and the POPE/POPG (3:1) lipid bilayer were assessed over the final 200 ns trajectories in CMD simulations.

	Contact Number	Protein Tilt Angle	ΔG_binding_	ΔG_elec_	ΔG_vdw_	ΔG_pols_	ΔG_npols_
CMD_M1	10,965 ± 671	29.0 ± 6.7	−1894 ± 130	−2420 ± 164	−619 ± 44	1224 ± 98	−79 ± 3
CMD_M2	11,759 ± 658	16.7 ± 7.0	−1933 ± 101	−2506 ± 128	−699 ± 38	1334 ± 158	−61 ± 4
CMD_M3	10,931 ± 660	30.9 ± 6.1	−1812 ± 143	−2045 ± 263	−664 ± 42	979 ± 212	−82 ± 4
CMD_M4	11,547 ± 612	11.5 ± 4.5	−1888 ± 115	−2359 ± 134	−661 ± 42	1193 ± 184	−62 ± 6
CMD_M5	10,317 ± 640	23.7 ± 7.0	−1741 ± 130	−1970 ± 163	−624 ± 39	930 ± 178	−78 ± 4

Furthermore, we determined the number of contacts between AapA1-28 protein and the POPE/POPG (3:1) bilayer during the last 200 ns of the CMD_M trajectories (presented in Table 2 and Appendix A). Regardless of K-23 interaction with the upper or bottom lipid membrane, no significant difference was observed in the number of contacts between AapA1-28 protein and the POPE/POPG (3:1) bilayer.

### 2.4. Structural Characteristics of the AapA1-28 Protein Based on CMD_M Simulations

To compare the structural characteristics of the AapA1-28 protein in the two distinct conformational states, we calculated the position of each residue in the five sets of CMD_M simulations. The distance between the backbone or side chain of each residue and membrane center was calculated and is depicted in Figure 6. Firstly, the CMD_M2 simulation exhibited similar structural characteristics of transmembrane residues 9–28 as observed in the CMD_M4 simulation. Similarly, the CMD_M1 simulation displayed comparable structural features of transmembrane residues 9–28 with CMD_M3 and CMD_M5 simulations. The distance between K-23 NH_3_^+^ group and the closest phosphate group of membrane is shown in Appendix A. There is a strong electrostatic interaction between the positively charged NH_3_^+^ of K-23 and the phosphate group of the membrane. Secondly, the primary difference between these two conformational states lies in the orientation of the K-23 side chain, whether it points towards the upper or lower membrane. The distance between K-23 NH_3_^+^ group and F-19 aromatic ring is shown in Appendix A. When the sidechain of K-23 residue points to the upper membrane, there is not only electrostatic interaction between K-23 and phosphate group of the membrane, but also cation–π interaction between K-23 and F-19.

## 3. Discussion

The AapA1 toxin is a small hydrophobic protein known for its toxic effect on *E. coli*. However, the AapA1-28 protein lacking the two C-terminal KR residues does not exhibit the same toxicity during *E. coli* growth. The C-terminal residues K-29 and R-30 are not only highly conserved across different *H. pylori* strains, but also play a critical role in the toxicity activity. Unfortunately, conventional bioinformatics prediction methods struggle to differentiate the antimicrobial/toxic properties of AapA1 and AapA1-28 (Table 3 presents the prediction results from various methods [24,25,26,27]). These methods yield different outcomes regarding the antimicrobial properties. The common sequence and structure characteristics of membrane-associated type I toxin protein were as follows: small hydrophobic protein containing less than 60 amino acids and have a putative α-helical transmembrane domain. However, when comparing the sequences and structural characteristics of membrane-associated type I toxin protein, it is hard to find the common mechanism of action. Additionally, the positive charge effect at the C-terminus is observed in three other type I TA toxins, namely, LdrD (with conserved KRK residues) [28], SprA1 (with KK residues) [29], and BsrG (with KKK residues) [30]. These toxins also exhibit a membrane toxic effect, indicating the vital role of these basic C-terminal residues in the toxicity process. Moreover, charged amino acids in type I TA toxin protein ZorO are important for its toxicity and how it affects remain unknown [31]. Therefore, it is important to analyze the structural characteristics of AapA1-28 in the *E. coli* membrane and compare them with those of the AapA1 toxin protein.

Numerous simulations results have consistently demonstrated that AapA1-28 protein has a greater tendency to insert into membranes at an angle rather than vertically. Additionally, the K-23 residue of AapA1-28 is more likely to interact with the upper membrane and forms cation–π interaction with F-19. Based on these findings, we hypothesize that the non-transmembrane conformation of AapA1-28 makes it challenging for the protein to form pores in the membrane. Conversely, in the case of the AapA1 protein, the polymers can insert vertically into the inner membrane of *E. coli*, forming pores that allow the passage of water or ions, thereby inducing toxicity (results in pending publication).

In this study, we employed 2D-MetaD simulations by using contact number between the K-23 residue and the phosphate group atoms as CVs. In protein–membrane 2D-MetaD simulations, the selection of appropriate CVs plays a crucial role in describing the translocation process. Although it is challenging to provide conclusive evidence that the chosen CVs effectively describe the protein translocation process, they offer a means to compare the free energy differences between the two distinct conformational states obtained from CMD_T1/2 simulations.

## 4. Conclusions

AapA1 and AapA1-28 protein are both inserted into bilayer membrane in an α-helix conformation, and it is difficult to distinguish and compare the structural mechanism of these proteins. Experimental method and conventional bioinformatics prediction methods struggle to differentiate the antimicrobial/toxic properties of AapA1 and AapA1-28 proteins. In this study, we elucidated the structural characteristics of the AapA1-28 protein within a POPE/POPG lipid bilayer membrane. Our findings revealed that, in contrast to the vertical insertion observed for the AapA1 protein, the AapA1-28 protein exhibited a higher propensity for insertion into the membrane at an angle. Additionally, we observed a greater likelihood of interaction between the K-23 residue of AapA1-28 and the upper membrane. Our results have shown that C-Terminal KR residues are essential for the transmembrane structure of the AapA1 toxin protein. AapA1 protein has different effects on *H. pylori* and *E. coli*. The toxic effect on *E. coli* of this protein can be transformed into potent antimicrobial drugs, and these findings contribute to development of new antimicrobial drugs.

## 5. Materials and Methods

### 5.1. MD Simulations for the AapA1-28 Tetramer-Containing Membrane Systems

We performed two sets of 5000 ns CMD simulations of the AapA1-28 tetramer-containing membrane systems (simulation details shown in Table 1). Each simulation contained four AapA1-28 proteins (sequence: MATKHGKNSWKTLYLKISFLGCKVVVLL, the sequence and structure characteristic of AapA1-28 are shown in Figure 3A,B). The initial conformations of the AapA1-28 monomer protein–membrane systems were derived by referring PWR and NMR experiments of full-length monomer AapA1 protein [4]. The AapA1-28 monomer protein was set as vertical transmembrane states and Ser9-Leu28 residues were set as a transmembrane region with an alpha-helix. Two sets of initial configurations of these tetrameric AapA1-28 protein–membrane systems were constructed (shown in Figure 1A,B). In the first conformation (Figure 1A), the protein tetramerized into a face-to-face tetramer, and four K-23 residue side chains faced the channel pore. In the second conformation (Figure 1B), the protein tetramerized into a face-to-back tetramer, and two K-23 residue side chains faced the bilayer.

A computer model for the *E. coli* inner membrane was built with 96 palmitoyloleoyl PE (POPE) lipids and 32 palmitoyloleoyl PG (POPG) lipids in the proportion 3:1. The simulation systems were created using CHARMM-GUI protocol (http://charmm-gui.org/) [32,33]. Then, we applied periodic boundary condition in a rectangle (the simulation box is 7.24 × 7.24 × 12 nm^3^ for tetramer AapA1-28-POPE/POPG [96:32]) and used TIP3P water model for solvation and 0.125 M NaCl for neutralization. The Chemistry at Harvard Molecular Mechanics (CHARMM) force field [21] was used to describe the proteins and membranes. Subsequently, extensive energy minimization was performed, followed by 5 ns equilibration simulations in both the NVT and NPT ensembles. All simulations were performed at a temperature of 310 K using the Nose−Hoover algorithm [34] and a pressure of 1 bar using the semi-isotropic Parrinello−Rahman method [35]. Particle-mesh Ewald method [36] was used to calculate the long-range electrostatic interactions with a grid spacing of 1.2 nm. The Lennard–Jones interaction was calculated using the force-based switching function with a cut-off of 1 nm. The bond length was constrained using the LINCS algorithm [37]. 

### 5.2. D-MetaD Simulations for Single AapA1-28-Containing Membrane Systems

We performed two sets of 1000 ns 2D-MetaD simulations for membrane systems containing monomeric AapA1-28 protein. The simulation details can be found in Table 1, and the initial conformations for these 2D-MetaD simulations, derived from CMD_T1/2 simulations, are illustrated in Figure 3C,D.

All 2D-MetaD simulations [38] were performed using GROMACS 2018 software package [39] patched with the PLUMED 2.5 plugin [40]. In order to compare the free energy differences between the two distinct conformational states from CMD_T1/2 simulations, we employed the total number of contacts (the CV is defined as coordination number in PLUMED) between K-23 residue and the phosphate group atoms in either the top or bottom leaflet of the membrane as the two collective variables (CVs) in these simulations. The details of CVs are shown in Table 1. The simulation box is approximately 6.29 × 6.29 × 15 nm^3^ for monomer AapA1-28-POPE/POPG [96:32]. The remaining simulation parameters were the same as above. The initial Gaussian height was set to 1 kJ/mol, and the width was set to 10 for the contact number. A bias factor of 10 was employed. The total time of the simulation was 1000 ns.

### 5.3. MD Simulations for Monomer AapA1-28-Containing Membrane Systems

We conducted five sets of 1000 ns CMD simulations for membrane systems incorporating monomeric AapA1-28. These simulations were initiated from the local energy minimum state obtained from the 2D-MetaD simulations, as described in Table 1. The simulation parameters remained the same as previously stated. All simulated systems were equilibrated for 800 ns, and subsequent 200 ns simulations were used for analysis.

## Figures and Tables

**Figure 1 toxins-15-00542-f001:**
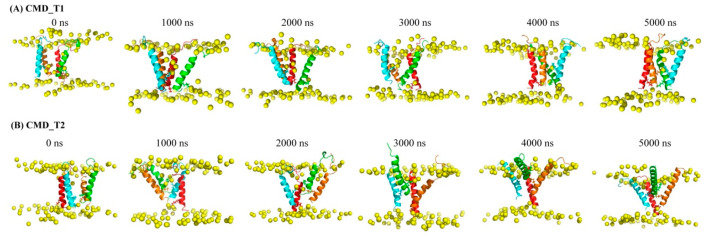
Snapshots of the tetrameric AapA1-28 protein–-membrane structure at various time points in (**A**) CMD_T1 simulation and (**B**) CMD_T2 simulation. The AapA1-28 proteins are represented by cartoons in different colors: red, green, orange, and cyan. The molecular dynamics (MD) simulation highlights the side chain orientation of the K-23 residue, which undergoes a transition from the lower membrane to the upper membrane. The K-23 residues are depicted as sticks. The phosphorus atoms of the lipid bilayer depicted in yellow ball.

**Figure 2 toxins-15-00542-f002:**
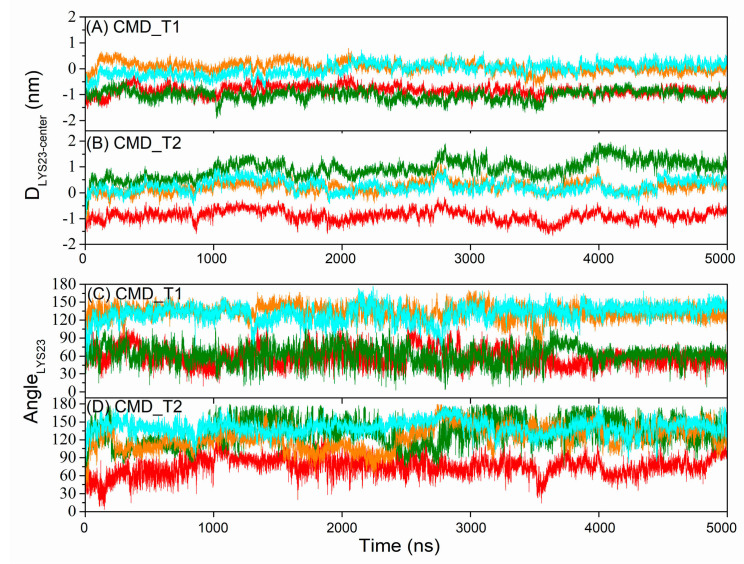
The time series of distance between K-23 side chain and membrane center in simulations CMD_T1 (**A**) and CMD_T2 (**B**). The time series of angle between K-23 side chain and membrane *Z*-axis in simulations CMD_T1 (**C**) and CMD_T2 (**D**). Red line: the first AapA1-28 protein; green line: the second AapA1-28 protein; orange line: the third AapA1-28 protein; cyan line: the fourth AapA1-28 protein.

**Figure 3 toxins-15-00542-f003:**
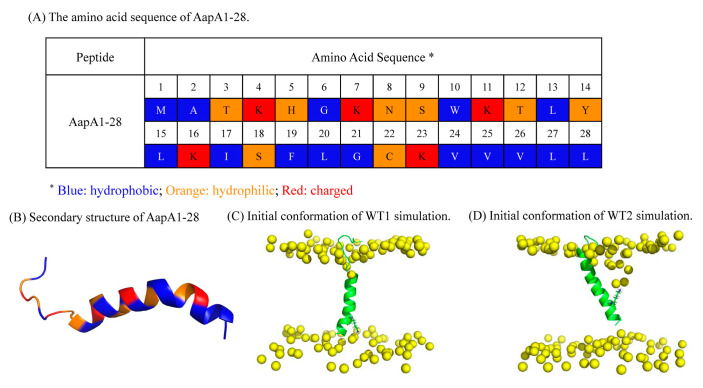
Structure of AapA1-28. (**A**) The amino acid sequence of AapA1-28 is presented, with different residue types represented by distinct colors: blue for hydrophobic residues, orange for hydrophilic residues, and red for charged residues. (**B**) The secondary structure of AapA1-28 is illustrated, highlighting its amphipathic α-helical configuration. The corresponding amino acid residues are color coded according to the scheme used in (**A**). (**C**,**D**) The initial conformation of the AapA1-28–POPE/POPG (3:1) system using 2D-MetaD simulations is shown, with the phosphorus atoms of the lipid bilayer and protein depicted in yellow and green, respectively.

**Figure 4 toxins-15-00542-f004:**
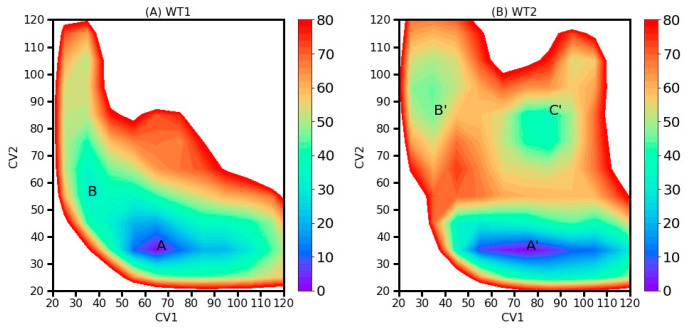
Two-dimensional free energy landscapes depicting the relationship between CV1 and CV2 for (**A**) WT1 and (**B**) WT2. The labeled local minima are denoted as A, B, A’, B’, and C’. The global minimum is assigned a free energy value of zero.

**Figure 5 toxins-15-00542-f005:**
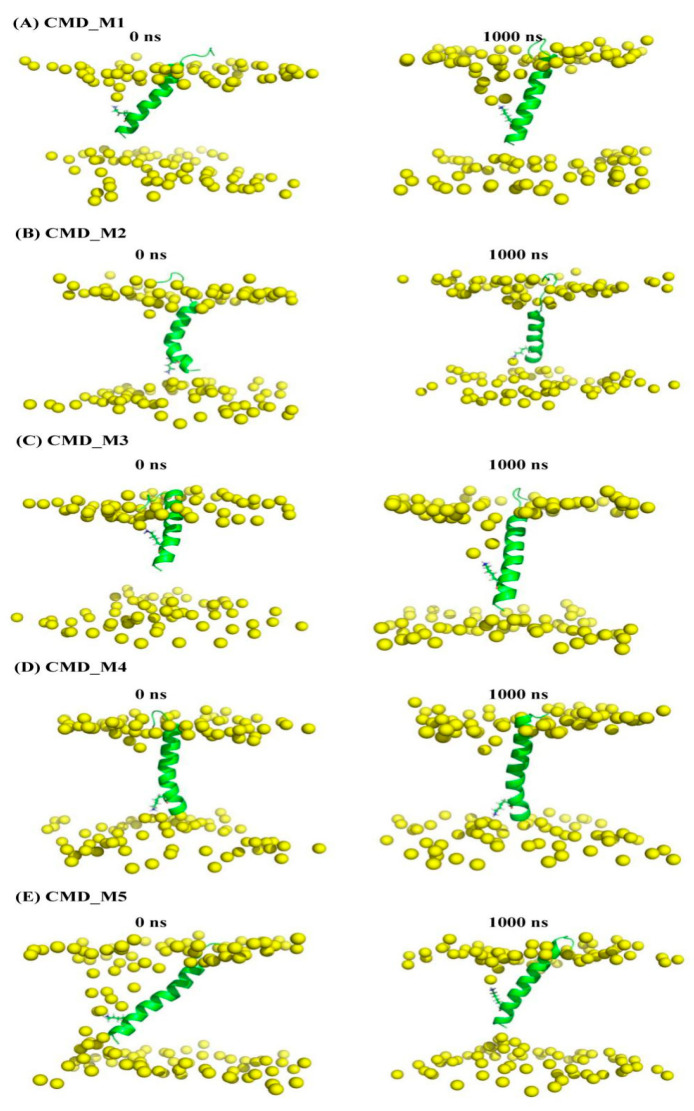
The representative conformation of local minima labeled as A, B, A’, B’, and C’ in Figure 4 as the initial conformation of CMD_M simulations. Snapshots of the AapA1-28 protein–membrane structure at 1000 ns. K-23 residues are shown as a stick. The phosphorus atoms of the lipid bilayer depicted in yellow ball.

**Figure 6 toxins-15-00542-f006:**
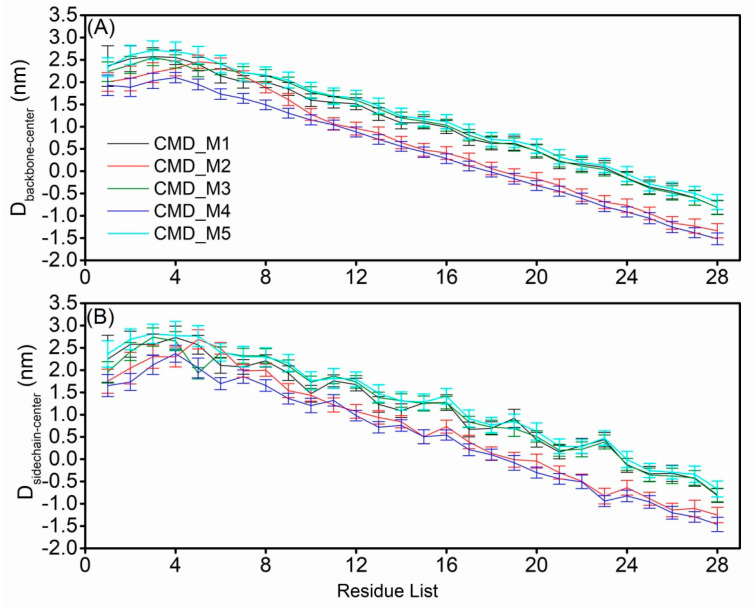
The average distance and variance (**A**) of each residue backbone to the membrane center and (**B**) of each residue side chain to the membrane center.

**Table 3 toxins-15-00542-t003:** Antimicrobial peptide prediction results.

Prediction Methods	AapA1 (MATKHGKNSWKTLYLKISFLGCKVVVLLKR)	AapA1-28 (MATKHGKNSWKTLYLKISFLGCKVVVLL)
AMP_Scanner	AMP (0.98)	AMP (0.88)
sAMPpred-GAT	AMP (0.70)	nonAMP (0.09)
AI4AMP	AMP (0.60)	nonAMP (0.46)
amPEPpy 1.0	nonAMP (0.29)	nonAMP (0.37)

## Data Availability

The data that support the findings of this study are available from the corresponding author upon reasonable request.

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
