# Peer review of "Effects of C-Terminal Lys-Arg Residue of AapA1 Protein on Toxicity and Structural Mechanism"

_toxins, 2023, doi:10.3390/toxins15090542_

Round 1

Reviewer 1 Report

H. pylori encodes a small toxin named AapA1 that kills cell by forming transmembrane pores. AapA1 contains two charged residues at the C-terminus, K and R, which upon removal inactivates the toxin. This paper aims at finding the reason for inactivation. By applying a number of sophisticated biophysical methods the authors come to the conclusion that the deletion derivative lacking K and R no longer assume a vertical orientation in the membrane but rather binds in an angled conformation in which K23 preferentially points to the upper membrane and forms a cation interaction with F19, in contrast to the full-length peptide in which K23 interacts with the lower membrane. Determination of the structural changes that inactivate the toxin is of great value since it helps to interpret the properties of other structurally similar toxins and of toxin mutant analyses. The data strongly indicate that this change of conformation causes inactivation of AapA1. 1. The source of the toxin and its deletion derivative, isolation procedure, purification, purity should be presented. 2. Indicate position of the phosphate group with which K23 interacts.

Author Response

  1. The source of the toxin and its deletion derivative, isolation procedure, purification, purity should be presented.

A:Thank you. In the revised manuscript, we added the description about the AapA1 toxin protein in introduction. We added “In order to assess the impact of C-terminal residues on the toxicity of AapA1, Korkut et al [4] produced specific constructs expressing truncated variant of the toxin that deleted for the last two Lys-Arg(KR) residues (named as AapA1-28 protein). The toxic effect of AapA1-28 protein was evaluated in liquid growth-media after over-expression of this protein during growth in E.Coli. While AapA1 displays the toxicity effect, the AapA1-28 protein does not exhibit any toxic effect on E.coli growth.”. In our simulations, the PDB data of AapA1-28 protein was obtained by deleting the PDB data file of AapA1.

  1. Indicate position of the phosphate group with which K23 interacts.

A:In the five sets of CMD_M simulations, the protein-membrane systems are inherently dynamic and sample a vast ensemble of conformation. It is difficult to label a single phosphate group that interacts with K23 residue. Hence, we calculated the distance between K-23 NH3+ group and the closest phosphate group of membrane and shown in Figure S2B.

Reviewer 2 Report

The manuscript addressed the Effects of C-Terminal KR Residue of AapA1 protein on toxicity 3 and structural mechanism. The manuscript is well-presented and generally OK.  However, the following points should be addressed by the authors before acceptance:

1- The title should be clear with identified abbreviations.

2- The significance/impact of the work should be mentioned at the end of the abstract and conclusions.

3- Latest key findings of previous related work should be outlined in the introduction.

4- In Figure 1, there is a table, authors should consider separation of both.

5- The scientific bacterial names should revised to be italic throughout entire manuscript.

It is generally OK.

Author Response

The title should be clear with identified abbreviations.

A:Thank you for pointing out this neglect, we have changed “KR” to “Lys-Arg” in the revised title. Thank you.

The significance/impact of the work should be mentioned at the end of the abstract and conclusions.

A:Thank you. In the revised manuscript, we added the significance of the work at the end of the abstract and conclusions.

Latest key findings of previous related work should be outlined in the introduction.

A:Thank you for pointing out this neglect, we added the latest key findings in the introduction.

In Figure 1, there is a table, authors should consider separation of both.

A:In order to demonstrate the sequence and structural characteristics of AapA1-28 protein, we put Figure 1A and 1B together and used to describe the polarity characteristics of each residue where each residue is represented with different colour.

  1. The scientific bacterial names should revised to be italic throughout entire manuscript.

A:Thank you for pointing out this neglect, we corrected it in the revised manuscript.

Round 2

Reviewer 2 Report

The authors have made all the changes requested to the satisfaction of the reviewer. I recommed the manuscript for publication.

Author Response

Thank you!